# Image-Based Dosimetry in Dogs and Cross-Reactivity with Human Tissues of IGF2R-Targeting Human Antibody

**DOI:** 10.3390/ph16070979

**Published:** 2023-07-08

**Authors:** Kevin J. H. Allen, Ohyun Kwon, Matthew R. Hutcheson, Joseph J. Grudzinski, Stuart M. Cain, Frederic A. Cruz, Remitha M. Vinayakamoorthy, Ying S. Sun, Lindsay Fairley, Chandra B. Prabaharan, Ryan Dickinson, Valerie MacDonald-Dickinson, Maruti Uppalapati, Bryan P. Bednarz, Ekaterina Dadachova

**Affiliations:** 1College of Pharmacy and Nutrition, University of Saskatchewan, Saskatoon, SK S7N 5E5, Canada; kja782@mail.usask.ca; 2Department of Medical Physics, University of Wisconsin-Madison, Madison, WI 53705, USA; okwon25@wisc.edu (O.K.); bbednarz2@wisc.edu (B.P.B.); 3Safety Resources, University of Saskatchewan, Saskatoon, SK S7N 5E5, Canada; matt.hutcheson@usask.ca; 4Department of Radiology, University of Wisconsin-Madison, Madison, WI 53705, USA; grudzinski@wisc.edu; 5adMare BioInnovations, Vancouver, BC V6T 1Z3, Canada; scain@admarebio.com (S.M.C.); ecruz@admarebio.com (F.A.C.); remithaaa@gmail.com (R.M.V.); ssun@admarebio.com (Y.S.S.); lfairley@admarebio.com (L.F.); 6Department of Pathology and Laboratory Medicine, College of Medicine, University of Saskatchewan, Saskatoon, SK S7N 5E5, Canada; chp347@mail.usask.ca (C.B.P.); maruti.uppalapati@usask.ca (M.U.); 7Department of Veterinary Pathology, Western College of Veterinary Medicine, University of Saskatchewan, Saskatoon, SK S7N 5B4, Canada; ryan.dickinson@usask.ca; 8Department of Small Animal Clinical Sciences, Western College of Veterinary Medicine, University of Saskatchewan, Saskatoon, SK S7N 5B4, Canada; valerie.macdonald@usask.ca

**Keywords:** IGF2R, osteosarcoma, image-based dosimetry RAPID, ^89^Zr, ^177^Lu, tissue cross-reactivity, PET/CT

## Abstract

Background: Osteosarcoma (OS) represents the most common primary bone tumor in humans and in companion dogs, being practically phenotypically identical. There is a need for effective treatments to extend the survival of patients with OS. Here, we examine the dosimetry in beagle dogs and cross-reactivity with human tissues of a novel human antibody, IF3, that targets the insulin growth factor receptor type 2 (IGF2R), which is overexpressed on OS cells, making it a candidate for radioimmunotherapy of OS. Methods: [^89^Zr]Zr-DFO-IF3 was injected into three healthy beagle dogs. PET/CT was conducted at 4, 24, 48, and 72 h. RAPID analysis was used to determine the dosimetry of [^177^Lu]Lu-CHXA”-IF3 for a clinical trial in companion dogs with OS. IF3 antibody was biotinylated, and a multitude of human tissues were assessed with immunohistochemistry. Results: PET/CT revealed that only the liver, bone marrow, and adrenal glands had high uptake. Clearance was initially through renal and hepatobiliary excretion in the first 72 h followed by primarily physical decay. RAPID analysis showed bone marrow to be the dose-limiting organ with a therapeutic range for ^177^Lu calculated to be 0.487–0.583 GBq. Immunohistochemistry demonstrated the absence of IGF2R expression on the surface of healthy human cells, thus suggesting that radioimmunotherapy with [^177^Lu]Lu-CHXA”-IF3 will be well tolerated. Conclusions: Image-based dosimetry has defined a safe therapeutic range for canine clinical trials, while immunohistochemistry has suggested that the antibody will not cross-react with healthy human tissues.

## 1. Introduction

Osteosarcoma (OS) represents the most common malignant primary bone tumor in dogs and humans and is responsible for 85–98% of malignancies forming in the skeleton in dogs [1] and 55% in children and adolescents [2]. Canine OS carries a poor prognosis with approximately 90% of affected dogs developing pulmonary metastases. The median survival time of dogs treated with amputation alone is only 4 months [3]. In human patients, the overall survival has similarly plateaued at approximately 70% with no meaningful improvement achieved within the last 25 years [2]. Thus, new therapeutic approaches to treating OS are urgently needed for both human and canine patients. The significance of the current study is the evaluation of a human antibody as a potential radioimmunotherapy agent for OS in vivo in canines and in vitro in human tissues.

A decade ago, insulin growth factor receptor type 2 (IGF2R) was identified as being overexpressed on the surface of all commercially available and human patient-derived OS cells [4]. These findings were later expanded to 34 consecutive cases of dogs with OS with all of them displaying some degree of IGF2R expression in the majority of the neoplastic osteoblasts [5]. As murine antibodies are not suitable for clinical trials in human patients because of immunogenicity issues, we created and molecularly characterized novel human antibodies to IGF2R that bind to human, canine, and murine forms of IGF2R [6]. The binding of these human antibodies to murine IGF2R would enable the initial evaluation of RIT efficacy and safety in mice with human OS xenografts, while the binding to canine IGF2R would afford a comparative oncology approach by treating OS-afflicted companion dogs with RIT. We have subsequently evaluated one of these antibodies, IF3, in severe combined immunodeficiency (SCID) mice bearing canine OS xenografts. In vivo single photon emission computed tomography/computed tomography (SPECT/CT) imaging revealed uptake of the IF3 antibody in the neoplastic cells of these xenografts [7]. When radiolabeled with therapeutic radionuclide ^177^Lu, the IF3 antibody significantly slowed down the growth of the xenograft tumors [7]. However, the biodistribution of IF3 in a larger animal model such as dogs and its potential cross-reactivity with human tissues remained to be investigated.

It is critical to study the safety and efficacy of theranostic agents that deliver therapeutic agents near organs at risk, particularly lymphoid organs (bone marrow, spleen, thymus, draining lymphatics). Furthermore, dosimetry calculations using canines should be more reliable for extrapolation to humans than mouse models. Here, we addressed the need for biodistribution and dosimetry data as well as for antibody cross-reactivity by performing image-based dosimetry estimations for [^177^Lu]Lu-CHXA”-IF3 antibody derived from the positron emission tomography/computed tomography (PET/CT) imaging of healthy beagle dogs with [^89^Zr]Zr-DFO-IF3 using PET/CT as well as tissue cross-reactivity evaluation of IF3 antibody with normal human tissues.

## 2. Results

### 2.1. [^89^Zr]Zr-DFO-IF3 Antibody Demonstrated Urinary and Hepatobiliary Excretion

Figure 1A,B shows the urinary and hepatobiliary excretion of [^89^Zr]Zr-DFO-IF3 after IV administration to the dogs. Most of the excretion took place within 73 h after administration, while after hours physical decay was primarily responsible for the elimination of ^89^Zr. The dose rate at 30 cm from the dog’s body surface fell from 12 µSv/h right after [^89^Zr]Zr-DFO-IF3 administration to 5 µSv/h at 48 h post administration (Figure 1C), which would allow the dogs to be released to their owners if companion dogs would be used in place of research dogs.

### 2.2. PET/CT Imaging Revealed Liver, Adrenals, and Bone Marrow as the Highest Uptake Organs

Figure 2, Figure 3 and Figure 4 display the maximum image projections (MIPs) of the dogs at four time points post [^89^Zr]Zr-DFO-IF3 administration and the pharmacokinetics information derived from those images. The organs with the highest [^89^Zr]Zr-DFO-IF3 uptake were, in decreasing order, the liver, adrenals, and marrow in the spine and the shoulders. The antibody was quickly cleared from the heart, and its retention in the testis and whole body was very low.

### 2.3. Image-Based Dosimetry Indicated the Bone Marrow as a Dose-Limiting Organ during RIT with [^177^Lu]Lu-CHXA”-IF3

The results of image-based dosimetry for [^177^Lu]Lu-CHXA”-IF3 are shown in Table 1. The [^177^Lu]Lu-CHXA”-IF3 activity for each dog was estimated with the goal of not exceeding a 3 Gy absorbed dose to the bone marrow, which is a dose-limiting organ for this radiopharmaceutical. The therapeutic activities of [^177^Lu]Lu-CHXA”-IF3 were found to be in the 0.487–0.563 GBq range in beagles.

### 2.4. Healthy Human Cells Do Not Express IGF2R on Their Surface

Biotinylated B-IF3 antibody produced weak to moderate cytoplasmic/cytoplasmic granule staining of occasional to frequent positive control 143B cells. B-IF3 antibody did not bind in a specific manner to K7M2 cells, which were utilized as the negative control. In its turn, the isotype control antibody, B-hIgG1, did not bind in a specific way to either IGF2R-positive 143B cells or IGF2R-negative K7M2 cells. Taken together, the B-IF3 binding to 143B cells, the absence of its binding to K7M2 cells, and the lack of B-hIgG1 isotype control antibody binding to either proved the specific and sensitive nature of the cell binding assay. Figure 5 displays the representative images of several major organs stained with B-IF3 and control B-hIgG1. Appendix A summarizes the intensity and frequency of the staining scores. Binding with B-IF3 was observed in the human tissue panel to the cytoplasm and/or cytoplasmic granules in the following organs:epithelial cells in the kidney (tubules), large intestine (colon) (mucosa), liver (hepatocytes), mammary gland (breast) (glands), pancreas (islets, acini, ducts), placenta (trophoblasts), skin (epidermis, sebaceous and sweat glands), small intestine (mucosa), and stomach (mucosa)precursor cells in the bone marrowmononuclear leukocytes in the esophagus, large intestine (colon) (gut-associated lymphoid tissue [GALT]), ovary, skin, and spleenKupffer cells in the liverspindle cells in the placenta (located in chorionic villi and, most likely, Hofbauer cells)reticulo-endothelial cells in the spleencells of glomerular tufts in the kidneymeningeal cells in the brain–cerebrum (falx cerebri)arachnoid cap cells in the brain–cerebrumneurons in the brain–cerebrum, small intestine (ganglia), and stomach (ganglia)

## 3. Discussion

As part of the preparation for a clinical trial of OS RIT with [^177^Lu]Lu-CHXA”-IF3 antibody in companion dogs with OS and subsequently in humans, we performed PET/CT imaging of healthy beagle dogs to enable image-based dosimetry estimations of [^177^Lu]Lu-CHXA”-IF3. In addition, the tissue cross-reactivity of IF3 antibody with normal human tissues was evaluated according to the requirements of the FDA Center for Biologics Evaluation and Research (CBER) document “Points to Consider in the Manufacture and Testing of Monoclonal Antibody Products for Human Use”.

The imaging performed over the period of 4 days confirmed hepatobiliary clearance of IF3 antibody, which is typical for antibodies as this is where antibodies are catabolized [8]. As the accumulation in the liver is not insignificant, it will need be taken into account for determining future patient dosing regimens to ensure that this organ will continue to remain non-dose limiting. IF3 did not concentrate in the spleen to any degree (Figure 2, Figure 3 and Figure 4) in contrast to previous murine work, confirming that IGF2R expression is specific for the spleens of Fox Chase SCID mice and, thus, will not be a targeting sink in canine or human patients. There was also no uptake in the normal bone; however, uptake was observed in what is believed to be bone marrow, as the majority of the activity is located within the center of the bone (spine, shoulders). This potentially can be attributed to the osteophilic properties of the ^89^Zr catabolites or [^89^Zr]Zr(Ox)_2_ [8,9]. As an increase in SUV in both the shoulder and spine ROI is observed over time (Figure 2, Figure 3 and Figure 4), it can support osteophilic catabolite accumulation as a source for this uptake. The RAPID platform used in this work to perform the image-based dosimetry estimations affords the estimation of absorbed doses, which will be delivered to normal organs and tumors [8,9,10,11]. Very recently, RAPID was used to calculate the doses of ^90^Y-small molecule NM600 for treatment of companion dogs with various advanced cancers [12]. For [^177^Lu]Lu-CHXA”-IF3, RAPID predicted the bone marrow to be a dose-limiting organ, which is often observed for the variety of radiotherapeutic antibodies in humans [13] and, based on this prediction, allowed for the estimation of the projected therapeutic doses of [^177^Lu]Lu-CHXA”-IF3 in companion dogs with OS.

It is generally accepted that the results from clinical trials in dogs in terms of safety and efficacy can be extrapolated to humans better than any other preclinical models. Companion canines with spontaneous tumors are attractive comparative models to humans for several reasons [14,15], including naturally occurring cancers, many with high recurrence and metastatic potential; strong genetic and molecular target similarities to human cancers; immune competence and native immuno-editing interactions between the tumor and host immune system; relevant tumor histologies with intratumoral and inter-individual heterogeneity; similar environmental carcinogen exposure to human cancers; and a more natural outbred population compared to inbred rodent model laboratory-derived canine populations. The inclusion of companion animals in the development and use of novel RIT agents also has advantages owing to their physical size and spatial distribution of tumors (primary and metastatic) and normal organs/tissues, which more closely mimics that in humans with cancer [16].

The analysis of the binding of biotinylated B-IF3 to normal human tissues revealed that all binding observed with B-IF3 was due to the IGF2R in the cytoplasm with none expressed on the surface of the normal cells. This is in contrast to the high expression of IGF2R on the surface of human and canine OS tumor cells [4,5,6,7]. As monoclonal antibodies cannot penetrate through the membranes of live cells, they cannot bind to their respective antigens located in the cytoplasm, and for this reason, such cytoplasmic antigens do not contribute to tissue cross-reactivity [17,18]. The observed staining with B-IF3 in mononuclear leukocytes, Kupffer cells in the liver, bone marrow precursor cells, and neurons was consistent with the reported expression of IGF2R [19,20,21]. This is an important observation, which means that the radiolabeled IF3 antibody will be binding in vivo only to the cancer cells that express IGF2R on their surface [4,5,6,7], while intracellular expression of IGF2R in normal tissue will remain invisible to the antibody, thus avoiding toxicity to normal tissues.

In the past, several bone-seeking radiopharmaceuticals were tried for treatment of OS; however, they cannot be of use for treatment of non-osseous metastases [22]. More recently, two clinical trials have been initiated that will investigate RIT of solid tumors with antibodies to IGF1R and HER-2 antigens labeled with alpha-emitters ^225^Ac and ^227^Th, respectively (NCT03746431 and NCT04147819) [23]. Such trials will generate useful information for developing the RIT approach to treatment of OS [23]. While these are also attractive targets in terms of OS, it is beneficial to have a wide variety of potential treatments giving clinicians more opportunity to treat individuals who may have different levels of antigen expression, allowing for a more personalized medical approach.

## 4. Materials and Methods

### 4.1. Animal Ethics and Approval

The study design was approved by the University of Saskatchewan’s Animal Research Ethics Board and adhered to the Canadian Council on Animal Care guidelines for humane animal use. The study was in compliance with appropriate ARRIVE guidelines.

### 4.2. Conjugation IF3 Antibody

IF3 human antibody to IGF2R [6] produced at the University of Saskatchewan was conjugated to the bifunctional chelator 1-(4-isothiocyanatophenyl)-3-[6,17-dihydroxy-7,10,18,21-tetraoxo-27-(N-acetylhydroxylamino)-6,11,17,22-tetraazaheptaeicosine] thiourea (*p*-SCN-Bn-DFO) (Macrocyclics, Plano, TX, USA) via modified literature methods [24]. In short, 800 μg of IF3 was exchanged into carbonate conjugation buffer, pH = 8.5, via spin filtration (30 kDa molecular weight cut off spin filter) and conjugated to the chelator using a 3-fold molar excess of *p*-SCN-Bn-DFO (2 mg/mL in DMSO) and incubated at 37 °C for 1.5 hrs. The IF3-DFO conjugate was then washed 10 times via spin filtration with 0.5 M HEPES buffer at 4 °C to remove excess *p*-SCN-Bn-DFO giving the IF3-DFO conjugate.

### 4.3. Labeling of IF3-DFO Conjugate

A total of 74 MBq of [^89^Zr]Zr(Ox)_2_ in 1 M oxalic acid (Sylvia Fedoruk Canadian Centre for Nuclear Innovation, Saskatoon, SK, Canada) was dissolved in 0.5 M HEPES buffer (which was previously passed through a Chelex-100 column to remove any trace metals) and neutralized using 1M Na_2_CO_3_. A total of 400 µg of IF3-DFO was then added to achieve a 0.185:1 MBq:µg specific activity. The reaction mixture was heated at 37° for 1 h and then quenched using 3 µL of 0.05 M DTPA solution; the percentage of radiolabeling yield was measured with instant thin layer chromatography (iTLC) (Agilent Technologies, Santa Clara, CA, USA) using 0.5 M EDTA as the eluant. The iTLC was cut in half and measured using a 2470 Wizard2 Gamma counter (Perkin Elmer, Waltham, MA, USA) calibrated for ^89^Zr emission spectra. Radiolabeling yields were calculated by dividing the counts per minute at the bottom half of the iTLC CPM by the total counts per minute (top + bottom), as the labeled antibody has a Rf = 0 vs. Rf = 1 for [^89^Zr]Zr-DTPA/EDTA. Radiolabeling yields were greater than 99%. The radiolabeled antibody was then exchanged into sterile phosphate-buffered saline (PBS) prior to injection.

### 4.4. PET/CT Imaging of Dogs

Animals were sedated with acepromazine 0.02 mg/kg combined with butorphanol 0.2 mg/kg IM with a top-up of a quarter to half of the original dose extra if necessary (if sedation was not adequate). A 20–22 G over-the-needle catheter was aseptically secured into the cephalic vein, and Ketamine 5 mg/kg combined with Midazolam 0.25 mg/kg was injected by IV in increments to achieve a surgical plane of anesthesia. The trachea was intubated with the dog in the sternal position with a cuffed endotracheal tube using a laryngoscope for visualization. The endotracheal tube was secured to the patient with k-ling. The endotracheal tube was attached to the anesthetic machine, which delivered isoflurane at an appropriate concentration to maintain a suitable anesthetic depth using a rebreathing system. Three doses of 11.1 MBq of purified labeled antibody were prepared just prior to injection of three beagle dogs: one female (F-1) weighing 10 kg and two males (M-1 and M-2) weighing 13 kg and 12 kg, respectively. Syringe radioactivity was measured before and after injection giving a total injected activity of 9.88 MBq, 10.0 MBq, and 9.47 MBq for F-1, M-1, and M-2, respectively. PET/CT scans were performed on a GE Discovery MI DR PET/CT scanner (GE healthcare, Waukesha, WI, USA) at 4, 24, 48, and 73 h post injection of [^89^Zr]Zr-DFO-IF3 (±0.6 h). The animals were immobilized in sternal recumbency after induction. A 256 × 256 matrix was used, DFOV 40 cm, acquired at 7 min/bed position (49–56 min total), immediately following a full body CT (2.5 mm/slice). The PET/CT images were registered and reconstructed automatically using GE’s Q. Clear algorithm, a Bayesian penalized-likelihood iterative image reconstruction, a β value of 550, and incorporating PSF and TOF corrections. A whole-body ROI was drawn around the subject (no excretion was confirmed prior) to confirm that the measured injected activity was consistent with the PET scanner and found to be ±10%.

### 4.5. Image-Based Dosimetry

Regions of interest (ROI) were drawn manually using the CT data for organ segmentation at each time point using 3D Slicer v5.0.3 (slicer.org) and exported as RTSS DICOM files for dosimetry analysis. The bone marrow dose was measured by drawing ROI on each shoulder and the spine as they showed a high amount of activity concentration. The dose rates were calculated at each of the imaging time points using a Monte Carlo (MC) dosimetry platform called RAPID (Radiopharmaceutical Assessment Platform for Internal Dosimetry) [25]. Following acquisition of the entire imaging series, the PET/CT imaging data were co-registered and resampled to the first time point of the CT image. Next, the activity concentration in each voxel was decay corrected from the imaging radionuclide (^89^Zr) to represent the therapeutic radionuclide (^177^Lu). The processed CT and PET imaging dataset was used in the MC simulations to define the geometry and source distributions at each time point, respectively. MC simulations were performed to determine the mean absorbed dose rate at each time point using the MC code Geant4 v9.6. A total of 8000 decays were simulated for each activity-rich voxel so that the uncertainty in average voxel dose rate was 1.07%. The mean absorbed dose rate of the corresponding voxels was integrated using a trapezoidal method to obtain the mean absorbed dose in each voxel. The estimation of activity concentration and standardized uptake values (SUV) were calculated in each ROI at each time point using voxel-level data. SUV was calculated using:(1)$$SUV=\frac{{\left[A_{voxel}\right]}_{@T_{n}}/V_{voxel}}{{\left[A_{injected}\right]}_{@T_{0}}\cdot e^{-\lambda T_{n}}/m_{subject}}$$
where ${\left[A_{voxel}\right]}_{@T_{n}}$ is the tracer radioactivity concentration in ROI voxels at *n*th time point, $V_{voxel}$ is the volume of the ROI voxels, ${\left[A_{injected}\right]}_{@T_{0}}$ is the injection activity, λ is the decay constant of imaging isotope, $T_{n}$ is the post-injection time, and $m_{subject}$ is the mass weight of the subject.

### 4.6. Tissue Cross-Reactivity Study with Biotinylated IF3 with Normal Human Tissues

Biotinylated IF3 (B-IF3) and biotinylated isotype control human hIgG1 (B-hIgG1) were generated by adMare BioInnovations (Vancouver, Canada). The human tissue cross-reactivity study was performed by Charles River (Morrisville, NC, USA) (Table 2). Cryosections of human OS 143B cells (ATCC, Manassas, VA, USA) were used as the positive control, while cryosections of murine IGF2R-negative OS K7M2 cells (ATCC, USA) served as the negative control. Positive and negative control cells were stored in a freezer set to maintain −65 °C or below.

A direct immunoperoxidase procedure was performed. Acetone-fixed cryosections were rinsed twice with PBS, pH 7.2. Slides were then incubated with an avidin solution for 15 min, rinsed once with PBS, incubated with a biotin solution for 15 min, and rinsed once with PBS. The slides were then treated for 20 min with a protein block designed to reduce nonspecific binding. The protein block was prepared as follows: PBS + 1% bovine serum albumin (BSA); 0.5% casein; 1.5% human gamma globulins (HGG); and 1 mg/mL heat aggregated human gamma globulins (HAHGG) (prepared by heating a 5 mg/mL solution to 63 °C for 20 min and then cooling to room temperature). Following treatment with the protein block, the biotinylated primary antibodies (B-IF3, B-hIgG1, or none [buffer alone as the assay control]) were applied to the slides at concentrations of 10 μg/mL for 1 h. Next, the slides were rinsed twice with PBS. Endogenous peroxidase was then quenched by incubation of the slides with the Dako peroxidase blocking solution for 5 min. Then, the slides were rinsed twice with PBS, treated with the ABC Elite reagent for 30 min, rinsed twice with PBS, and then treated with DAB for 4 min as a substrate for the peroxidase reaction. All slides were rinsed with tap water, counterstained, dehydrated, and mounted. PBS + 1% BSA served as the diluent for all antibodies and the ABC Elite reagent.

After staining, the slides were visualized under light microscopy for immunopathology. Each stained cell type or tissue element was identified, the subcellular (or extracellular) location of the staining was recorded, and the intensity (strength) of staining (Table 3) was assigned for each slide. The frequency of cell type staining (Table 4) was also assigned to provide the approximate percentage of cells of that particular cell type or tissue element with staining.

## 5. Conclusions

In this study, we performed image-based dosimetry for ^89^Zr/^177^Lu-labeled IF3 human antibody to IGF2R and evaluated IF3 binding to normal human tissues. The results of the study demonstrate that IF3 has a typical human antibody biodistribution profile and does not cross-react with any normal human tissues, thus informing future radioimmunotherapy studies in canine and human patients with OS.

## 6. Patents

ED and MU are co-inventors on the Provisional US Patent Application “Antibodies to IGF2R and Methods” filed on 30 March 2021.

## Figures and Tables

**Figure 1 pharmaceuticals-16-00979-f001:**
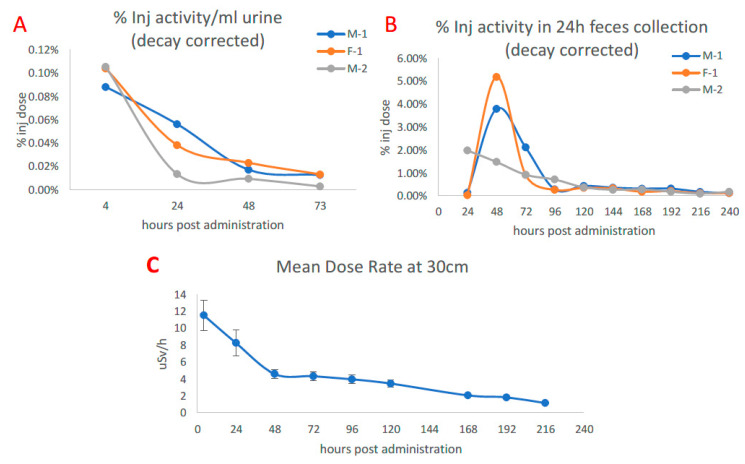
Excretion of [^89^Zr]Zr-DFO-IF3 after administration to beagle dogs and dose rate at 30 cm from the canine body surface. (**A**) Urine measurement via PET. (**B**) Feces collected over the course of monitoring. (**C**) Average dose measured at 30 cm from the dog over isolation period.

**Figure 2 pharmaceuticals-16-00979-f002:**
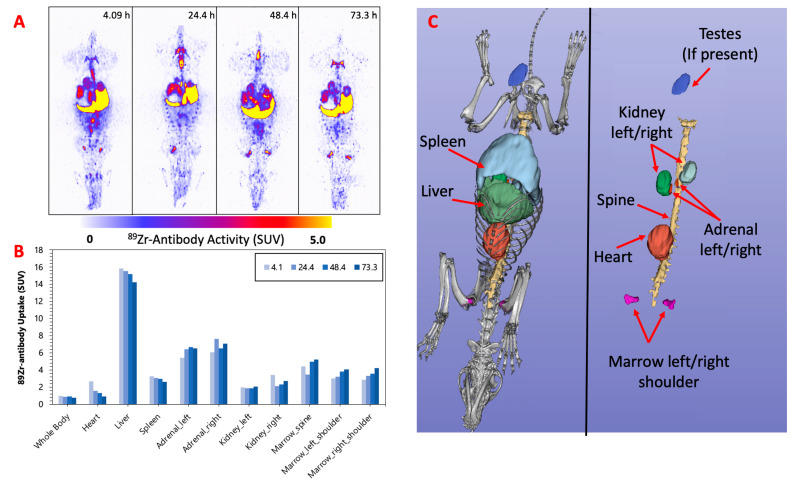
(**A**) F-1—MIPs of PET image volumes. (**B**) Uptake of [^89^Zr]Zr-DFO-IF3 (SUV) in ROIs as a function of scan times; (**C**) CT-derived position of dog organs.

**Figure 3 pharmaceuticals-16-00979-f003:**
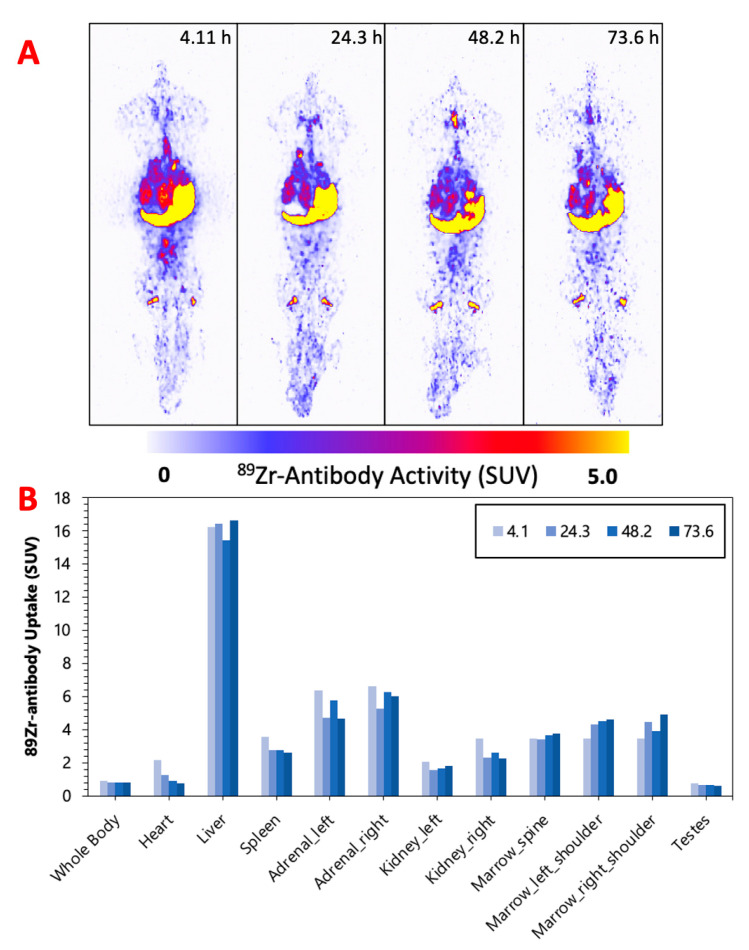
(**A**) M-1—MIPs of PET image volumes. (**B**) Uptake of [^89^Zr]Zr-DFO-IF3 (SUV) in ROIs as a function of scan times.

**Figure 4 pharmaceuticals-16-00979-f004:**
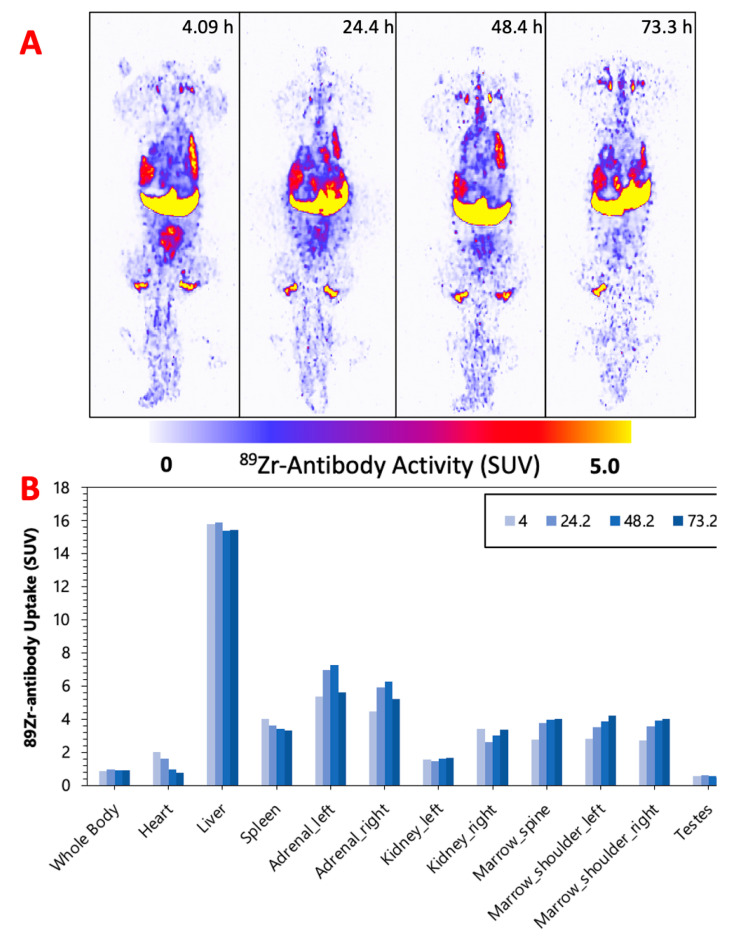
(**A**) M-2—MIPs of PET image volumes. (**B**) Uptake of [^89^Zr]Zr-DFO-IF3 (SUV) in ROIs as a function of scan times.

**Figure 5 pharmaceuticals-16-00979-f005:**
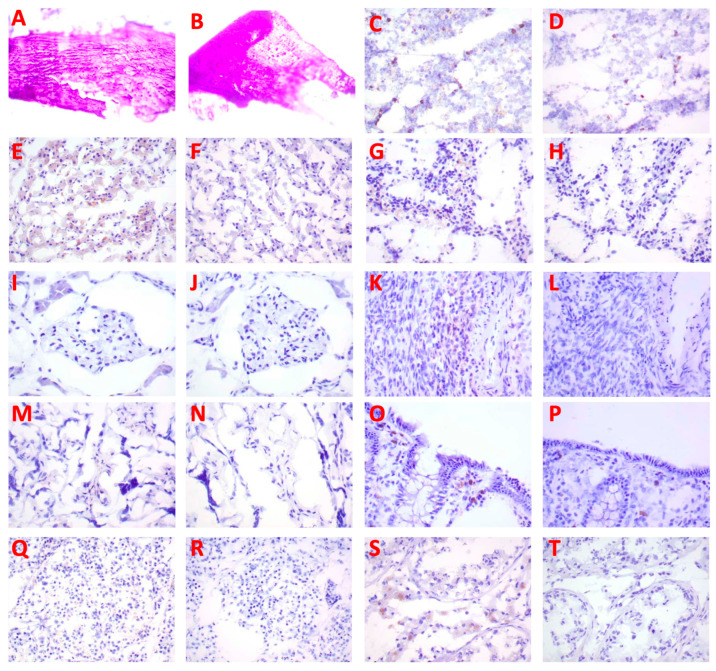
IHC of human tissues (40×) with biotinylated IF3 antibody and human isotype matching control hIgG1. Bone: (**A**) no staining with B-IF3, (**B**) no staining with B-hIgG1. Bone marrow: (**C**) positive staining of precursor cells with B-IF3, (**D**) no staining with B-hIgG1. Liver: (**E**) positive staining of Kupffer cells and hepatocytes with B-IF3, (**F**) no staining with B-hIgG1. Spleen: (**G**) positive staining of reticuloendothelial cells with B-IF3, (**H**) no staining with B-hIgG1. Kidney: (**I**) positive staining of glomerular tuft cells with B-IF3, (**J**) no staining with B-hIgG1. Ovary: (**K**) positive staining of mononuclear leukocytes with B-IF3, (**L**) no staining with B-hIgG1. Placenta: (**M**) positive staining of spindloid cells with B-IF3, (**N**) no staining with B-hIgG1. Large intestine: (**O**) positive staining of epithelial cells with B-IF3, (**P**) no staining with B-hIgG1. Pancreas: (**Q**) positive staining of islet cells with B-IF3, (**R**) no staining with B-hIgG1. Human testis: (**S**) positive staining of germinal epithelial cells and interstitial cells with B-IF3, (**T**) no staining with B- hIgG1.

**Table 1 pharmaceuticals-16-00979-t001:** Results of dosimetry calculations for [^177^Lu]Lu-CHXA”-IF3 for individual dogs.

	F-1		M-1		M-2	
ROI	Rx Dose (Gy/GBq)	0.487 GBq [^177^Lu]Lu-IF3 (Gy)	Rx Dose (Gy/GBq)	0.555 GBq [^177^Lu]Lu-IF3 (Gy)	Rx Dose (Gy/GBq)	0.563 GBq [^177^Lu]Lu-IF3 (Gy)
Heart	2.28	1.11	1.43	0.79	1.66	0.94
Liver	26.89	13.10	23.49	13.04	24.34	13.71
Spleen	5.14	2.51	3.93	2.18	5.31	2.99
Adrenal_left	12.35	6.01	7.42	4.12	9.74	5.48
Adrenal_right	13.52	6.59	9.18	5.10	9.08	5.11
Kidney_left	3.97	1.93	2.75	1.53	2.76	1.56
Kidney_right	5.32	2.59	3.69	2.05	5.41	3.05
Marrow_spine	7.87	3.83	4.44	2.47	5.00	2.82
Marrow_left_shoulder	6.16	3.00	5.40	3.00	5.33	3.00
Marrow_right_shoulder	6.27	3.05	5.67	3.15	5.00	2.82
Testes	-	-	0.90	0.50	0.80	0.45

**Table 2 pharmaceuticals-16-00979-t002:** Human Tissue (Normal) from One Individual.

Tissues		
Bone	Breast (mammary gland)	Ovary
Bladder (urinary)	Gastrointestinal (GI) Tract ^b^	Pancreas
Blood Vessels (endothelium) ^a^	Heart	Placenta
Bone Marrow	Kidney (glomerulus, tubule)	Skin
Brain—cerebrum	Liver	Spleen
	Lung	

^a^ Evaluated from all tissues where present. ^b^ Includes esophagus, large intestine/colon, small intestine, and stomach (including underlying smooth muscle).

**Table 3 pharmaceuticals-16-00979-t003:** Scoring scale for the intensity of IHC staining.

Staining Intensity	
Score	Result
Neg	Negative (no stained cells)
±	Equivocal (very faint stain)
1+	Weak (light stain)
2+	Moderate (light–medium stain)
3+	Strong (medium stain)
4+	Intense (dark stain)

**Table 4 pharmaceuticals-16-00979-t004:** Scoring scale for the frequency of IHC staining.

Staining Frequency	
Score	Result
Neg	Negative (no stained cells)
Very rare	<1% stained cells of a particular cell type or tissue element
Rare	1–5% stained cells of a particular cell type or tissue element
Rare to Occasional	>5–25% stained cells of a particular cell type or tissue element
Occasional	>25–50% stained cells of a particular cell type or tissue element
Occasional to Frequent	>50–75% stained cells of a particular cell type or tissue element
Frequent	>75–100% stained cells of a particular cell type or tissue element

## Data Availability

Data is contained within the article and Appendix A.

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
