# Peer review of "Image-Based Dosimetry in Dogs and Cross-Reactivity with Human Tissues of IGF2R-Targeting Human Antibody"

_pharmaceuticals, 2023, doi:10.3390/ph16070979_

Round 1

Reviewer 1 Report

The authors evaluated a potential of radioimmunotherapy of osteosarcoma targeting IGF2R. Development of new agents for target radionuclide therapy will be considered important work.

I have some comment.

1.      The main problem is the high liver accumulation of the antibody. I would expect this to be a specific accumulation, since there was little accumulation in the heart and high in the liver at 4 hours after administration. Is expression level of IGF2R high in dogs? Please show the data and compare with human. If the expression level in the liver of dogs is much higher than in humans, then it would not be a problem, but if not, high hepatic accumulation is also expected in humans. In the discussion section, the authors mentioned that liver accumulation is not a problem, but if accumulation level is too high, liver damage would be a concern. In addition, if the antibody accumulates in the liver early in administration, it is expected that blood clearance will be fast and tumor accumulation level will be low. In such a case, the dose would need to be increased, and there would be more concern about side effects of liver.

2.      Biodistribution of dogs and mice were much difference in your studies. This difference may involve differences in IGF2R expression patterns as well as body size. Thus, I wonder if human results are predictable from dog results? Please provide any evidence for this.

3.      Introduction is long and unorganized. Please revise it with more focus.

Minor

Page 3, line 122: “89” should be superscripted. No period is needed.

Fig 2-4: It is difficult to understand the position of each organ in the PET image. So, a photo or sketch of the dog is required. It would be better to label each accumulation.

The authors described that PET scans were performed 73 h after injection. But results of excretion analysis were 72 h after injection. Is that correct?

Author Response

The authors evaluated a potential of radioimmunotherapy of osteosarcoma targeting IGF2R. Development of new agents for target radionuclide therapy will be considered important work. – Response: We would like to thank the Reviewer for his/her favorable opinion about our work.

I have some comment.

  1. The main problem is the high liver accumulation of the antibody. I would expect this to be a specific accumulation, since there was little accumulation in the heart and high in the liver at 4 hours after administration. Is expression level of IGF2R high in dogs? Please show the data and compare with human. If the expression level in the liver of dogs is much higher than in humans, then it would not be a problem, but if not, high hepatic accumulation is also expected in humans. In the discussion section, the authors mentioned that liver accumulation is not a problem, but if accumulation level is too high, liver damage would be a concern. In addition, if the antibody accumulates in the liver early in administration, it is expected that blood clearance will be fast and tumor accumulation level will be low. In such a case, the dose would need to be increased, and there would be more concern about side effects of liver. – Response: There is no evidence in literature that liver in dogs expresses IGF2R extracellularly and as our own data for a human liver shows (please see Fig. 5E) – there is no extracellular expression of IGF2R in human liver either. This means that IF3 antibody does not accumulate in the liver specifically. As IF3 antibody clears from the blood really fast (Fig. 2-4) – its catabolism in the liver also starts early. Fast clearance of IF3 from the blood could be actually an advantage to reduce toxicity to the bone marrow which is dose limiting organ according to the dosimetry calculations (Table 1). Importantly, DFO bifunctional chelating agent used for labeling the antibodies with 89Zr, is known to be lipophilic and might add to the liver accumulation of the antibody. During radioimmunotherapy clinical trials another bifunctional chelating agent – a macrocyclic DOTA will be used for labeling the antibody with 177Lu, as DFO is only used for 89Zr. DOTA is a widely used chelating agent in the clinic with very low lipophilicity, so we expect less uptake of the antibody in liver in both dogs and human during the clinical trials. 
  2. Biodistribution of dogs and mice were much difference in your studies. This difference may involve differences in IGF2R expression patterns as well as body size. Thus, I wonder if human results are predictable from dog results? Please provide any evidence for this. – Response: We are not stating that the human results could be predicted from the dogs trial. The only prediction is relative to anti-tumor efficacy of the radiolabeled antibody. All other information such as pharmacokinetics and safety can only be derived from a carefully designed Phase 1 clinical trial in human patients. 
  3. Introduction is long and unorganized. Please revise it with more focus. – Response: We have reorganized and considerably shorten the Introduction.

 Minor

Page 3, line 122: “89” should be superscripted. No period is needed. – Response: Has been corrected.

Fig 2-4: It is difficult to understand the position of each organ in the PET image. So, a photo or sketch of the dog is required. It would be better to label each accumulation. – Response: We have added an image of dog’s organs derived from CT (Fig. 2C).

The authors described that PET scans were performed 73 h after injection. But results of excretion analysis were 72 h after injection. Is that correct? – Response: We apologize for this typo and we have corrected both Fig. 1 and the text to 73 hrs.

Reviewer 2 Report

Dear Author(s),

Thank you for your efforts, and please make the following changes to your manuscript in order for it to be publishable and acceptable:

1. The article's title should be revised so that it more accurately conveys the objective of the current study.

2. The title should not exceed twenty words, as the number of words is one of the distinguishing features of a distinguished research title.

3. The abstract of the study is well-written and does not require revision. The study's introduction is excessively lengthy and must be condensed.

4. The following must be considered when making corrections:

 a. The introduction to the study should contain no more than three paragraphs.

 b. The significance of the current study should be stated in the first paragraph of the introduction.

 c. The second paragraph of the study's introduction should describe the knowledge gaps that the study intends to fill.

 d. The final paragraph of the introduction to the study should describe the problem of the current research and how to address it within the context of the study's objectives.

5. Materials and Methods must come before Results. Also, the Discussion section must follow the Results section. Please rectify this mistake.

6. I did not locate any clarification in the Materials and Methods section regarding the statistical method used to analyze the study's results. I hope the author(s) will also address this flaw.

7. In the conclusion, include a sentence describing whether the current research problem has been resolved; in other words, whether the study's objectives have been met.

8. A considerable number of references in the present study are out-of-date and should be replaced with more recent sources. In addition, the number of references is excessive for the requirements of this paper. Therefore, I recommend that the author(s) use only references from 2023 and at least five years prior, while minimizing unnecessary references.

//Good Luck//

The current manuscript required minor English language editing.

Author Response

Thank you for your efforts, and please make the following changes to your manuscript in order for it to be publishable and acceptable: - Response: We would like to thank the Reviewer for his/her favorable review.

  1. The article's title should be revised so that it more accurately conveys the objective of the current study. – Response: We have revised the title.
  2. The title should not exceed twenty words, as the number of words is one of the distinguishing features of a distinguished research title. - Response: We have revised and shortened the title.
  3. The abstract of the study is well-written and does not require revision. The study's introduction is excessively lengthy and must be condensed. – Response: We have shortened the Introduction to three paragraphs.
  4. The following must be considered when making corrections:
  5. The introduction to the study should contain no more than three paragraphs. – Response: Done.
  6. The significance of the current study should be stated in the first paragraph of the introduction. - Response: Done.
  7. The second paragraph of the study's introduction should describe the knowledge gaps that the study intends to fill. - Response: Done.
  8. The final paragraph of the introduction to the study should describe the problem of the current research and how to address it within the context of the study's objectives. - Response: Done.

  1. Materials and Methods must come before Results. Also, the Discussion section must follow the Results section. Please rectify this mistake. - Response: The format of the Journal requires the Results section to be placed after Introduction.
  2. I did not locate any clarification in the Materials and Methods section regarding the statistical method used to analyze the study's results. I hope the author(s) will also address this flaw. – Response: We did not perform any statistical analysis as with 3 dogs in the study (2 males and 1 female) the study does not have any statistical power.
  3. In the conclusion, include a sentence describing whether the current research problem has been resolved; in other words, whether the study's objectives have been met. – Response: We have changed the Conclusion accordingly.
  4. A considerable number of references in the present study are out-of-date and should be replaced with more recent sources. In addition, the number of references is excessive for the requirements of this paper. Therefore, I recommend that the author(s) use only references from 2023 and at least five years prior, while minimizing unnecessary references. – Response: We have reduced the number of references from 49 to 25 and have replaced many older references with the newer ones.

Round 2

Reviewer 1 Report

The authors well revised the manuscript.